# Role of Adhesion G Protein-Coupled Receptors in Immune Dysfunction and Disorder

**DOI:** 10.3390/ijms24065499

**Published:** 2023-03-13

**Authors:** Wen-Yi Tseng, Martin Stacey, Hsi-Hsien Lin

**Affiliations:** 1Division of Rheumatology, Allergy, and Immunology, Chang Gung Memorial Hospital-Keelung, Keelung 20401, Taiwan; 2Department of Medicine, College of Medicine, Chang Gung University, Taoyuan 33302, Taiwan; 3Whole-Genome Research Core Laboratory of Human Diseases, Chang Gung Memorial Hospital, Keelung 20401, Taiwan; 4Faculty of Biological Sciences, School of Molecular and Cellular Biology, University of Leeds, Leeds LS2 9JT, UK; 5Department of Anatomic Pathology, Chang Gung Memorial Hospital-Linkou, Taoyuan 33305, Taiwan; 6Graduate School of Biomedical Sciences, College of Medicine, Chang Gung University, Taoyuan 33302, Taiwan; 7Department of Microbiology and Immunology, College of Medicine, Chang Gung University, Taoyuan 33302, Taiwan

**Keywords:** adhesion GPCRs, immune dysfunction, immune disorder, ligand, signaling

## Abstract

Disorders of the immune system, including immunodeficiency, immuno-malignancy, and (auto)inflammatory, autoimmune, and allergic diseases, have a great impact on a host’s health. Cellular communication mediated through cell surface receptors, among different cell types and between cell and microenvironment, plays a critical role in immune responses. Selective members of the adhesion G protein-coupled receptor (aGPCR) family are expressed differentially in diverse immune cell types and have been implicated recently in unique immune dysfunctions and disorders in part due to their dual cell adhesion and signaling roles. Here, we discuss the molecular and functional characteristics of distinctive immune aGPCRs and their physiopathological roles in the immune system.

## 1. Introduction

Adhesion G protein-coupled receptors (aGPCRs) make up the second-largest GPCR subfamily which comprises a total of 33 different members in humans [1]. Several distinctive structural and functional characteristics of aGPCRs set them apart from other GPCRs. The N-terminal extracellular region of aGPCRs is markedly large and comprises many different cell-adhesive protein motifs arranged in tandem or in mixed combination. These protein motifs, including the epidermal growth factor (EGF)-, immunoglobulin (Ig)-, lectin-, leucine-rich repeat (LRR)-, pentraxin (PTX)-, thrombospondin (TSP)-like, and hormone-binding domains, all have well-known protein–protein interaction and cellular adhesion functions [1]. A hallmark GPCR autoproteolysis-inducing (GAIN) domain is located immediately after the cell-adhesive protein motifs and before the signature seven-transmembrane (7TM) domain (Figure 1). These structural features stipulate bilateral extracellular adhesion and intracellular signaling functions for aGPCRs. Indeed, diverse ligands/interacting partners of endogenous and exogenous origins, including soluble, cell surface, and cell matrix proteins, carbohydrates, lipids, and microbial products have been identified for aGPCRs [1]. In addition, both G protein-dependent and -independent signaling pathways were known to be induced by aGPCRs [2,3].

For the majority of aGPCRs, a peculiar auto-proteolytic modification takes place post-translationally at a conserved HL/T(S) sequence of the consensus GPCR proteolytic site (GPS) within the GAIN domain through nucleophilic reactions [4,5]. Interestingly, these reactions occur in the endoplasmic reticulum during receptor biosynthesis and result in a non-covalent two-subunit complex consisting of an extracellular N-terminal fragment (NTF) and a 7TM C-terminal fragment (CTF) [4] (Figure 1). Intriguingly, GPS auto-proteolysis is a feature singular only to aGPCRs and a handful of multi-transmembrane proteins such as polycystin-1, suggesting a novel role in regulating receptor activation and function [6].

Undoubtedly, one of the most accepted aGPCR activation mechanisms is the tethered agonism model which designates that receptor activation is triggered by the exposure of the N-terminal tethered peptide (called Stachel) of CTF following the shedding of NTF induced by ligand binding. The newly unveiled Stachel peptide then bends and docks into its own 7TM region, causing conformational changes and intracellular signaling [7,8]. Thus, GPS proteolysis and the dissociation/dislocation of NTF from CTF are thought as a prerequisite for the tethered agonistic activation of most aGPCRs [9]. This unusual activation mechanism has recently marked several aGPCRs as functional mechanosensing receptors in many biological systems [10,11]. Apart from this, GPS cleavage-independent and NTF-CTF dissociation-independent activation mechanisms have also been identified, indicating versatile activation modes for aGPCRs. As a result, aGPCRs have been associated with diverse physiopathological processes including immune responses [12].

Indeed, cell surface receptors such as GPCRs play a prominent role in the immune system not only as the signaling molecules responding to changes in the extracellular milieu but also as unique markers of distinct immune cell populations. To date, a distinctive group of aGPCRs, mostly selective members of the ADGRB, ADGRE, and ADGRG subfamily, have been found to be expressed differentially in multifarious immune cell types [13,14] (Table 1). Importantly, functional studies have identified novel roles of these immune aGPCRs in innate and adaptive immune responses as well as the differentiation and development of immune cells. Moreover, deregulated aGPCR expression and/or activity due to somatic mutations or irregular transcriptional/translational control have been linked to unique immune dysfunctions and disorders (Table 1). Herein, we summarize the current understanding of these immune aGPCRs and discuss their physiopathological roles in the immune system.

## 2. ADGRB1/BAI1

### 2.1. Molecular and Functional Characterization of ADGRB1/BAI1

ADGRB1/BAI1 belongs to the ADGRB subfamily whose three members were named brain-specific angiogenesis inhibitors (BAIs) based on their ability to inhibit angiogenesis and tumor formation. All BAIs possess a unique NTF consisting of four to five thrombospondin type 1 repeats (TSRs) followed by a single hormone-binding domain (HBD) before the GAIN domain (Figure 2) [15,16,17]. Noticeably among aGPCRs, a relatively long C-terminal tail comprising a PDZ domain-binding motif critical for signal transduction is present in BAIs.

BAI1 was identified initially as a potential brain-specific p53-regulated target, but later studies showed that its expression is not restricted to the brain nor strictly p53-dependent [18,19]. Interestingly, unique proteolytic modification of BAI1-NTF produced distinct soluble ectodomains, named vasculostatin 120 and vasculostatin 40, that possess potent antiangiogenic activities via a CD36-dependent mechanism [20,21,22]. BAI1 is highly expressed in various brain cells including neurons, astrocytes, and microglia. Expression was also detected in bone marrow, spleen, colon, and testis as well as macrophages and skeletal myoblasts [16]. To date, several endogenous and exogenous ligands/interacting partners including lipopolysaccharide (LPS), phosphatidylserine (PS), complement C1q protein, CD36, αvβ5 integrin, and Reticulon-4 receptors (RTN4Rs) have been identified for BAI1 [21,23,24,25,26,27].

Apart from its well-reported inhibitory roles in in vitro and in vivo angiogenesis and tumor growth, a critical function in synaptogenesis, spatial learning, and neuronal development has also been established for BAI1 [27,28,29]. Importantly, accumulating evidence has emerged for the involvement of BAI1 in the regulation of immune responses and immune system disorders [13,14].

### 2.2. Physiopathological role of ADGRB1/BAI1 in the Immune System

Swift clearance of apoptotic cells is key to tissue homeostasis and aberrant accumulation of uncleared dead cells often leads to chronic inflammation and tissue damage even immune disorder, such as autoimmune diseases [30]. Through the interaction of its TSRs with PS that is enriched on the outer membrane leaflet of apoptotic cells, BAI1 was identified as a novel phagocytic receptor for apoptotic cells both in macrophages and non-professional phagocytes, such as fibroblasts and epithelial cells [23]. Upon the binding of PS, the cytoplasmic tail of BAI1 interacts with the conserved engulfment signaling molecules, engulfment and cell motility 1 (ELMO1) and Dock180, to induce the activation of Rac small GTPase for apoptotic cell clearance [23]. Subsequent studies by Das et al. showed that BAI1 is also involved in the uptake of apoptotic gastric epithelial cells by gastric phagocytes following *Helicobacter pylori* infection [31].

Concomitantly, Lee et al. demonstrated that *Bai1*-deficient cells are less able to engulf apoptotic cells, leading to the accumulation of uncleared apoptotic corpses in vivo during tissue injury [32]. In a colonic inflammation model of dextran sodium sulfate (DSS)-mediated colitis, progressively down-regulated Bai1 expression was noted in colonic tissues and gut epithelial cells after DSS administration. Moreover, *Bai1*-deficient animals were more susceptible to DSS-induced colitis, showing augmented disease severity and decreased survival accompanied by excess uncleared cell corpses and increased inflammatory cytokines within the colonic epithelium. By contrast, attenuated DSS-induced colonic inflammation was found in transgenic mice overexpressing BAI1 specifically in colonic epithelial cells but not in myeloid cells. Finally, transgenic mice expressing a signaling-deficient BAI1 mutant were unable to attenuate DSS-induced colitis. These results indicate that enhancing apoptotic cell engulfment by BAI1 signaling in colonic epithelial cells might have a beneficial effect in dampening inflammatory responses in colonic inflammatory disorders, such as Crohn’s disease and ulcerative colitis [32].

More recently, apoptotic cell-induced BAI1 signaling in macrophages was found to upregulate ATP-binding cassette transporter 1 (ABCA1), which is a cholesterol efflux transporter important for the biogenesis of high-density lipoprotein (HDL) [33]. As a result, *Bai1*-deficient animals had reduced serum levels of total cholesterol, HDL, low-density lipoprotein (LDL), and triglycerides due to the attenuation of ABCA1 upregulation in response to apoptotic cells. Conversely, Bai1-overexpressing transgenic mice displayed higher ratios of HDL/total cholesterol and HDL/LDL [33]. As elevated HDL-cholesterol levels are known as a dominant negative risk factor of coronary artery disease [34], the BAI1-ELMO1-Rac1 pathway may be a potential target for developing therapeutics to combat cardiovascular diseases.

In line with previous findings, BAI1 was shown to mediate the engulfment of dying neurons by microglia in embryonic zebrafish brains. Mazaheri et al. revealed that zebrafish microglial BAI1 is specifically involved in the formation and transport of phagosomes around dying neurons while another PS receptor, TIM-4, is needed to stabilize the phagosome structure for the efficient clearance of dying neurons [35]. As many brain pathologies, such as Alzheimer’s disease, are closely associated with impaired microglial phagocytosis, BAI1-mediated microglial neuronal removal might provide important insights into the pathogenic mechanisms of certain brain disorders.

In the innate immune system, BAI1 was one of the first aGPCRs to be identified as a pattern recognition receptor that interacted with LPS via its TSRs to stimulate gram-negative bacteria engulfment by macrophages [24]. Again, BAI1-mediated bacterial phagocytosis is dependent on the ELMO/Dock/Rac signaling axis, which promotes inflammatory response and microbicidal activity by upregulating the production of pro-inflammatory cytokine TNF and reactive oxygen species (ROS) [24,36]. Consequently, *Bai1*-deficient mice were more susceptible to peritoneal infection due to impaired bacterial clearance. BAI1, therefore, plays a role in innate immunity against bacterial infection. In addition, BAI1 was reported to be involved in anti-viral responses in a glioma-bearing mouse model similarly by promoting TNF production in macrophages/microglia [37].

More recently, another BAI1-associated signaling pathway was linked to the pathogenesis of T-cell acute lymphoblastic leukemia (T-ALL), a malignant disease of T progenitor cells [38]. The coexistence of PHF6 and JAK3 mutations in T-ALL patients was frequently noted, but little is known of the mechanism whereby PHF6 mutations promoted JAK3-driven T-cell leukemia. Yuan et al. showed that the BAI1-MDM2-p53 pathway is inhibited by PHF6 deficiency in mutant JAK3-driven T-ALL. BAI1 has been shown previously to prevent MDM2-mediated p53 ubiquitination, hence its degradation [39]. PHF6, a transcriptional regulator, was able to up-regulate BAI1 to maintain p53 levels and thus suppress leukemia development. However, this inhibitory effect is lost in PHF6-deficient T-ALL driven by mutant JAK3. Critically, combined therapy with JAK3 and MDM2 inhibitors significantly reduced the leukemia burden of these T-ALL in vivo, suggesting a therapeutic role of the BAI1-MDM2-p53 pathway for T-ALL patients with PHF6 and JAK3 comutation [38].

Despite the numerous reports of BAI1 as a macrophage phagocytic receptor, a recent paper by Hsiao et al. paradoxically pointed out the inability to detect its gene and protein expression in multiple monocyte/macrophage cell subsets [40]. Perhaps, BAI1 expression in myeloid cell lineages is under strict regulation and only induced in certain conditions. Taken together, BAI1 is a multifunctional aGPCR with diverse roles in the innate as well as adaptive immune systems, including bacterial and viral infection, tissue homeostasis, chronic tissue inflammation, and T-cell leukemia.

## 3. ADGRE1/EMR1 (Emr1, F4/80)

### 3.1. Molecular and Functional Characterization of ADGRE1/EMR1

The mouse macrophage-specific F4/80 antigen (Ag) is the target and the namesake of the F4/80 monoclonal antibody (mAb) developed in rats against thioglycollate-elicited peritoneal macrophages [41]. Extensive expressional analyses have established it as a robust surface marker of restricted murine myeloid cell subsets, including monocytes, most tissue macrophages, eosinophils, and some dendritic cells (DCs) [42,43,44]. As such, the F4/80 Ag has been used widely as an excellent marker of mouse tissue macrophages, including tumor-associated macrophages [45,46].

Molecular cloning and phylogenetic analyses have established F4/80 as the murine ortholog of the human *ADGRE1/EMR1* gene that encodes the first member of the ADGRE subfamily in which E represents the EGF motif (Figure 2) [47,48]. F4/80 (Adgre1/Emr1) and EMR1 possess seven and six tandem EGF motifs, respectively, before the GAIN domain [49,50,51]. Interestingly, different from other ADGRE members, both F4/80 and EMR1 are un-cleaved single-chain receptors due to the presence of an uncanonical GPS sequence.

### 3.2. Physiopathological Role of ADGRE1/EMR1 in the Immune System

In a study to investigate the possible role of F4/80 in the antibacterial response of macrophages, Warschkau et al. used the F4/80 mAb in an in vitro infection model of the facultative intracellular pathogen [52]. It was known that during the heat-killed *Listeria monocytogenes* (HKL) infection of whole splenic cells from severe combined immunodeficiency (SCID) mice, the crosstalk between macrophages and NK cells was important in producing significant levels of IFN-γ, which is critical for restricting intracellular bacterial growth. Interestingly, the presence of the F4/80 mAb in the culture down-regulated the macrophage production of TNF and IL-12 that were needed to activate NK cells to produce IFN-γ. Of note, this inhibitory effect of F4/80 mAb was evident only when both macrophage and NK cell types were in close contact [52]. It was hence concluded that F4/80 likely interacts with a novel ligand on the NK cell surface to induce cytokine release.

Subsequently, two different F4/80 knock-out mouse strains were developed for the study of its biological function [53,54]. Unexpectedly, both strains of *F4/80*-deficient mice were healthy and phenotypically normal with no apparent defect in the differentiation and function of tissue macrophages. This indicated that the F4/80 molecule is not essential for the in vivo development of mouse monocyte/macrophage. Nevertheless, a role of F4/80 in the induction of peripheral immune tolerance was established using in vitro and in vivo models of anterior chamber-associated immune deviation (ACAID) [54]. Specifically, we showed that *F4/80*-deficient mice fail to generate functional efferent CD8^+^ regulatory T (Treg) cells to suppress Ag-specific delayed-type hypersensitivity (DTH) responses following Ag inoculation into the anterior chamber. Consistently, *F4/80*-deficient mice were equally unable to produce an efficient immune tolerance response in a low-dose oral tolerance model. Furthermore, it was found that the presence of F4/80^+^ APCs is absolutely required for the generation of efferent Treg cells in an in vitro ACAID model via an F4/80 Ag-dependent mechanism. Finally, the adoptive transfer of F4/80^+^ APCs into *F4/80*-deficient mice restored peripheral tolerance in both ACAID and low-dose oral tolerance models, indicating a critical role for the F4/80 receptor in the generation of efferent CD8^+^ T reg cells [54]. The mechanism of F4/80-mediated Ag-specific peripheral immune tolerance remains undefined but likely involves a cellular ligand/interacting partner.

Of interest, unlike F4/80, EMR1 was shown to be expressed predominantly in human eosinophilic granulocytes [55,56]. Importantly, Legrand et al. demonstrated that in vivo administration of an afucosylated EMR1-specific mAb efficiently depletes eosinophils in monkeys and dramatically induces in vitro killing of eosinophils via NK-mediated cytotoxicity. Thus, EMR1 seems to be a specific target for eosinophil depletion in the treatment of eosinophilic disorders [56]. Nevertheless, Waddell et al. later carried out RNA-Seq analyses of ADGRE1 expression in various macrophage populations of 8 different mammalian species and detected its consistent expression in macrophages in all species tested, suggesting a conserved expression pattern of EMR1 in macrophages [57].

In conclusion, EMR1 (Emr1, F4/80) represents a restricted aGPCR marker of monocyte/macrophage and eosinophilic granulocyte with a role in the anti-bacterial response, peripheral immune tolerance, and the control of eosinophilic disorders.

## 4. ADGRE2/EMR2

### 4.1. Molecular and Functional Characterization of ADGRE2/EMR2

ADGRE2/EMR2 is an F4/80-like human aGPCR with five EGF motifs, however, unlike *F4/80* (*Emr1*), *EMR2* orthologs are completely absent in rodents. Extensive alternative splicing of *EMR2* transcripts in exons encoding the EGF motifs resulted in at least four receptor isoforms including EMR2(EGF1,2), EMR2(EGF1,2,5), EMR2(EGF1,2,3,5), and EMR2(EGF1–5) [58]. Intriguingly, amino acid sequences of the five EGF motifs of EMR2 are 97% identical to those of ADGRE5/CD97 whereas its 7TM region shares ~90% sequence identity with that of ADGRE3/EMR3 [59,60]. This suggests that *EMR2* is a genetic chimera of *CD97* and *EMR3*, most probably due to gene duplication and conversion [61]. Therefore, EMR2 likely interacts with similar/same cellular ligand(s) of CD97 but induces signaling activities analogous to those of EMR3. Indeed, the full-length EMR2(EGF1–5) variant shares a glycosaminoglycan (GAG) ligand, chondroitin sulfate B/dermatan sulfate, with CD97(EGF1–5) [62]. On the other hand, EMR2(EGF1,2,5) binds to CD55 (decay-accelerating factor, DAF) with a 10-fold weaker affinity of CD55-CD97(EGF1,2,5) interaction (Figure 2) [63].

Similar to F4/80, EMR2 is expressed restrictedly in myeloid cells, including monocytes, macrophages, myeloid DCs, and granulocytes [64]. EMR2 expression is differentially regulated during in vitro differentiation of macrophages and DCs, while the strongest in vivo EMR2 protein signal is detected in CD16^+^ blood monocytes and BDCA-3^+^ myeloid DCs [64,65]. Foamy macrophages in atherosclerotic vessels and splenic Gaucher cells are strongly EMR2-positive, but foam cells of multiple sclerosis brains express very little, if any, EMR2 [66]. Likewise, abundant EMR2 expression is detected in subsets of macrophages and neutrophils within inflamed tissues [65]. Altogether, these expressional data indicate strongly a modulatory role of EMR2 in the innate immune functions of human myeloid cells.

### 4.2. Physiopathological Role of ADGRE2/EMR2 in the Immune System

In human neutrophils, the ligation and activation of EMR2 by an NTF-specific 2A1 mAb significantly potentiated cell activation and recruitment in response to a panel of inflammatory stimuli [67]. Moreover, EMR2 activation up-regulated the production of multiple cytokines and suppressed LPS-induced prolonged neutrophil survival, suggesting that EMR2 activation induces a priming effect for neutrophil activation [68]. Lewis et al. identified increased EMR2 expression in blood neutrophils of systemic inflammatory response syndrome (SIRS) patients and found a positive association of the percentage of EMR2^+^ neutrophils with the extent of organ failure, making EMR2 a possible neutrophil biomarker for SIRS [69]. Similarly, blood neutrophils of liver cirrhosis patients with infection expressed higher levels of EMR2, which correlated strongly with the disease severity and predicted the overall mortality of patients [70].

The identification of dermatan sulphate as a specific GAG ligand of EMR2 lead to the finding that the interaction of this ligand-receptor pair might play a role in recruiting macrophages into the inflamed synovium tissues of rheumatoid arthritis (RA) [71]. Critically, a missense EMR2-C492Y variant was identified recently as the disease protein responsible for vibratory urticaria (VU), an autosomal dominant autoinflammatory dermal allergy caused by vibration or repetitive stretching [72]. Mechanically, the noncovalent NTF-CTF association of the C492Y variant was weakened and became easily separated upon vibratory stimulation in the presence of dermatan sulphate or 2A1. The vibratory activation of EMR2-C492Y triggered massive mast cell activation and elevated histamine discharge, hence the dermal allergic response [72]. Naranjo et al. later reported that EMR2-C492Y activation in mast cells induced specific activation of PLC-β, PI3K, ERK1/2, and a transient increase in cytosolic calcium via a Gβγ-, Gα_q/11_- and Gα_i/o_-independent mechanism [73]. The role of EMR2 in mast cell-associated pathology was further validated in another monogenic immune disorder called hereditary α-tryptasemia, which is characterized by clinical manifestations such as VU and dysautonomia due to increased α-tryptase expression. Le et al. showed that EMR2 on mast cells was susceptible to proteolytic cleavage and activation by α/β-tryptase, leading to cell degranulation when vibrated while adherent to dermatan sulfate [74]. EMR2 hence is one of the few mechanosensing receptors to be linked directly to a specific autoinflammatory disorder.

We have investigated the role of EMR2 activation and signaling in macrophage biology and showed that cross-linking of EMR2 by 2A1 induced the mobilization of its NTF and CTF into lipid rafts. This in turn led to receptor signaling and the production of inflammatory cytokines [75]. We subsequently demonstrated that 2A1-induced EMR2 activation facilitated the differentiation and inflammatory activation of human monocytic cells, inducing the production of IL-8, TNF-α, and MMP-9 via the Gα_16_/Akt/PLC-β/MAPK/NF-κB pathway [76]. More recently, we showed that EMR2 activation plays a critical role in triggering the activation (2nd) signal for NLRP3 inflammasome activation in both THP-1 monocytic cells and primary monocytes, producing significant levels of IL-1β and IL-18 in the presence of pathogen-associated molecular patterns (PAMPs), such as LPS [77].

Concurrently, Irmscher et al. identified a complement regulatory protein, factor H-related protein 1 (FHR1), as a novel soluble ligand of EMR2 and showed that FHR1-EMR2 interaction activates human monocytes to induce NLRP3 inflammasome activation in the presence of normal human serum [78]. Specifically, FHR1 interacts with EMR2 through its short consensus repeat (SCR) 3–5 domains on one hand and binds to malondialdehyde-modified low-density lipoproteins (MDA-LDL) on necrotic cell surfaces via its SCR1-2 domains on the other hand. Hence, FHR1 functions essentially as a molecular bridge linking both necrotic cells and monocytes to trigger pro-inflammatory responses through EMR2 via a Gβγ-dependent PLC signaling pathway. As such, FHR1 was frequently found adherent to necrotic areas of atherosclerotic plaques and the necrotic glomerular sites of anti-neutrophil cytoplasmic antibody-associated vasculitis (AAV) patients. Moreover, the serum FHR1 levels of AAV patients correlated negatively with the glomerular filtration rate, but positively with the inflammatory status [78]. These results altogether highlight a critical role for FHR1 in promoting sterile inflammation and auto-inflammatory diseases by activating the EMR2 receptor on human monocytes.

In short, EMR2 seems to function as an activation receptor in various human myeloid cell types including neutrophil, mast cell, and macrophage with a role in (auto)inflammatory responses and unique immunopathology such as VU and α-tryptasemia.

## 5. ADGRE5/CD97

### 5.1. Molecular and Functional Characterization of ADGRE5/CD97

As mentioned, the structural organization of human ADGRE5/CD97 protein is highly similar to that of EMR2 and three human receptor isoforms, namely CD97(EGF1,2,5), CD97(EGF1,2,3,5), and CD97(EGF1–5) are identified as a result of alternative splicing [79]. To date, four CD97-interacting ligands recognized by its various extracellular domains were identified. These are CD55 [80], chondroitin sulfate B/dermatan sulfate [62], integrin α5β1 [81], and CD90 (Thy-1) [82]. In addition, Ward et al. recently showed a specific *cis*-partnership between CD97 and lysophosphatidic acid receptor 1 (LPAR1), another GPCR, that likely form a heterodimer via their 7TM domains (Figure 2) [83].

Despite the strong similarity in protein sequences, the *CD97* gene is evolutionarily conserved in all vertebrate genomes, unlike *EMR2* [84]. Moreover, CD97 is expressed in similar patterns in human and mouse cells alike, further suggesting functional conservation. CD97 was identified initially as a surface marker of activated lymphocytes [85], but its expression was later detected in hematopoietic stem and progenitor cells (HSPCs) and various cell types of lymphoid and myeloid lineages [86,87]. In addition, many normal and malignant non-hematopoietic cell types such as smooth muscle cells and colorectal carcinoma cells also express CD97 [88,89,90].

### 5.2. Physiopathological Role of ADGRE5/CD97 in the Immune System

CD97 has long been implicated in leukocyte adhesion/trafficking and immune effector functions due to its unique expressional and molecular characteristics [80]. As such, Kwakkenbos et al. showed that CD97(EGF1–5) on activated T cells facilitated T cell-B cell interaction by binding to dermatan sulfate on circulating B cells [91]. In psoriatic skin lesions, the interaction of upregulated CD97 on polymorphonuclear cells with CD90 on activated endothelial cells was thought to engage in regulating leukocyte trafficking during inflammatory responses [82]. Likewise, CD97-CD55 interaction was functionally implicated in cellular adhesion and trafficking in autoimmune disorders, such as RA and multiple sclerosis [92,93].

To investigate the functional role of CD97 in immunity, Hamann’s group developed and used specific mAbs against its extracellular domains in several knock-out models of innate and adaptive immune challenges. Their results showed that CD97 mAb administration caused retarded neutrophil migration to sites of inflammation in vivo and thus impaired anti-bacterial host defense and survival [94], attenuated inflammatory responses to experimental arthritis [95], and reduced stem cell mobilization from bone marrow [96]. However, no obvious effects on several Ag-specific adaptive immune disease models, such as delayed-type hypersensitivity or experimental autoimmune encephalomyelitis, were detected using these mAbs [97]. A follow-up study showed that the CD97 mAb eliminated granulocytes in the bone marrow and blood via an Fc receptor-dependent but complement activation-independent mechanism [98].

Despite these findings, no apparent phenotypes were found in two different strains of *Cd97* knock-out mice at steady states, except for a mild granulocytosis [99,100]. Interestingly, granulocyte accumulation at sites of inflammation was normal in one *Cd97*-deficient mouse strain following *S. pneumoniae*-mediated lung infection [99]. By contrast, a different *Cd97*-null mouse strain showed improved anti-bacterial host response to systemic infection by *Listeria monocytogenes*, apparently due to enhanced granulocyte accumulation in infected tissues [100]. Nevertheless, similar in vitro and in vivo migratory properties were identified in the Cd97 null and wild-type granulocytes. It was concluded that CD97 is dispensable for normal leukocyte trafficking, but is involved in peripheral granulocyte homeostasis by a yet unknown mechanism [100]. Interestingly, animals that were deficient in Cd55 showed a comparable granulocytosis phenotype with approximately two-fold more granulocytes in circulation. Concomitantly, enhanced anti-bacterial response and improved host survival were found in *Cd55*-deficient mice infected with *S. pneumonia* [101]. Likewise, reduced arthritis activity was similarly detected in *Cd97*-deficient and *Cd55*-deficient mice in two different experimental models of RA [102]. Altogether, CD97 likely plays a role in granulocytic homeostasis and anti-bacterial response in part via binding to CD55.

Regarding the functional role of CD97 in other leukocyte types, Capasso et al. showed that in vitro co-stimulation of CD3 and CD55, either by mAbs or soluble CD97, lead to increased proliferation, activation, and enhanced IL-10 and granulocyte-macrophage colony-stimulating factor (GM-CSF) secretion of peripheral blood CD4^+^ T cells without interfering with CD55-mediated complement regulation [103]. The same group later reported that co-stimulation of human naïve CD4^+^ T cells via CD55-CD97 interaction drives the activation and expansion of a minor T regulatory type 1 (Tr1) cell population that suppresses T cell functions by an IL-10–dependent mechanism [104]. These results suggested strongly a functional role for CD97-CD55 interaction in regulating T cell functions.

Intriguingly, surface CD97 levels were significantly increased on CD55-deficient leukocytes when compared with wild-type cells. Furthermore, surface CD97 was rapidly downregulated to normal levels following the adoptive transfer of CD55-deficient leukocytes into wild-type mice due to cell-cell contact in a shear stress-dependent manner [105]. Thus, CD97, like EMR2, functions as a mechanosensing receptor with a role in leukocyte biology via binding to CD55 [106]. Indeed, a recent paper by Liu et al. confirmed that CD97 plays an essential role in the correct localization of type-2 conventional DCs (cDC2s) in the bridging channels between the white pulp and red pulp areas of the spleen by interacting with CD55-expressing red blood cells. As such, CD97 is involved in promoting the homeostasis and Ag-presenting functions of cDC2s in confined splenic niches via mechanosensing red blood cells through CD55 [107].

Recently, a role for CD97 in maintaining the stability of the immunological synapse (IS) between DCs and T cells was established by Cerny et al. who showed that the SteD effector protein of *Salmonella enterica* targets surface CD97 on bacteria-infected DCs for protein degradation, resulting in impaired DC-T cell interaction and T cell activation due to reduced CD97 levels in the IS [108]. It was not known at present whether CD55 is the interacting partner of CD97 in the IS. Coincidently, Xu et al. investigated the immune modulatory capacity of specific chiral nanoparticles and found that these nanoparticles targeted Cd97 and Emr1 on mouse bone marrow-derived DCs (BMDCs) for endocytosis thereby triggering inflammasome activation via activating mechanosensitive potassium-efflux channels, eventually leading to enhanced BMDC maturation and Ag-specific immune functions [109]. In conclusion, CD97 plays an immunoregulatory role in adaptive immunity by regulating the expansion, homeostasis, and immune activation of T cells and DCs.

A possible role for CD97 in regulating hematopoiesis has been suggested by its specific expression in HSPCs and up-regulated expression in the majority of primary acute myeloid leukemia (AML) specimens [110,111]. Additionally, CD97 was identified as a possible MYC-target gene specifically expressed in Burkitt lymphoma (BL), but not in diffuse large B-cell lymphoma (DLBCL) [112]. Importantly, higher CD97 expression levels were associated with poorer overall survival of AML patients, while CD97-knockdown negatively modulated cellular adhesion, migration, and FMS-like tyrosine kinase 3 (FLT3) gene expression in some AML cell lines [110,113]. Finally, Martin et al. showed that CD97 is an important functional regulator of AML stem cells by promoting the proliferation, survival, and undifferentiated phenotype of leukemic blast cells via multiple signaling pathways [114].

Altogether, CD97 is an aGPCR critically involved in the proliferation, homeostasis, and effector functions of diverse myeloid and lymphoid cells as well as leukemic stem cells (LSCs).

## 6. ADGRG1/GPR56

### 6.1. Molecular and Functional Characterization of ADGRG1/GPR56

The *ADGRG1/GPR56* gene juxtaposes with that of *ADGRG3/GPR97* and *ADGRG5/GPR114* on chromosome 16q21. Thus, these three genes are highly homologous and all are expressed in leukocytes. As the disease-associated molecule responsible for bilateral frontoparietal polymicrogyria (BFPP), GPR56 is best known for its role in cortical and neuronal development [115]. Structural analysis has revealed a unique pentraxin/laminin/neurexin sex-hormone-binding globulin-like (PLL) domain and GAIN domain in GPR56-NTF [116]. As with BAI1 and CD97, multiple ligands/binding partners of GPR56 have been identified, including tetraspanins CD9/CD81, tissue transglutaminase (transglutaminase-2, TG2), collagen III, laminin, heparin/heparan sulfate, progastrin, amino acid L-Phe, and phosphatidylserine (PS) (Figure 2) [117,118,119,120,121,122,123,124].

### 6.2. Physiopathological Role of ADGRG1/GPR56 in the Immune System

Other than the nervous system, protein expressional analysis of human leukocyte populations has identified GPR56 as a marker of pan-cytotoxic leukocytes, including NK and cytotoxic CD8^+^, CD4^+^, and γδT cells [125,126]. Moreover, increased GPR56 expression was identified in virus-specific CD8^+^ T cells following cytomegalovirus infection, while a reduced spontaneous and SDF-1-stimulated trans-well migratory response was displayed by GPR56-expressing NK-92 cells [126]. These findings suggest a role of GPR56 in inhibiting lymphocyte migration. Consistently, investigation of GPR56-expressing NK-92 cells and GPR56-null primary NK cells isolated from BFPP patients showed that GPR56 reduces cellular degranulation, production of inflammatory cytokines and cytolytic proteins, and cytolytic activity in association with CD81 *in cis* [127]. GPR56, therefore, is an NK immune checkpoint inhibitory receptor which regulates its cytotoxic effector functions negatively.

The recent investigation of GPR56 expression in T cells has further defined its functional association with unique effector T cell subsets. Truong et al. showed that GPR56 is progressively expressed in human CD4^+^ memory T cell subpopulations that produce the highest pro-inflammatory cytokines TNF and IFN-γ [128]. Immunoprofiling of synovial CD4^+^ T cells of anti-citrullinated protein antibodies (ACPA)^+^ and ACPA^−^ RA patients using the single cell sequencing technology identified specific GPR56 expression in the CXCL13^high^ peripheral helper T (T_PH_) cell subset that contains most clonally expanded T cells in the inflamed joints [129]. Similar findings were reported by Lutter et al. who showed that GPR56 expression defines a specific suppressive CD161^+^CXCL13^+^ Treg cell subset in patients with juvenile idiopathic arthritis (JIA) [130].

In line with these results, we identified increased soluble GPR56 (sGPR56) in the serum of RA, but not systemic lupus erythematosus, Sjogren’s syndrome, or ankylosing spondylitis patients [131]. Moreover, the elevated serum sGPR56 levels correlated positively with several RA biomarkers, including rheumatoid factor and TNF, suggesting that sGPR56 might be a novel serum biomarker for active RA. Coincidentally, Zeng et al. recently showed that increased GPR56^+^ cytotoxic T lymphocyte fractions in active RA patients correlate positively with the disease severity/progression [132]. GPR56 expression in NK cells was regulated predominantly by the transcription factor Hobit [127], however, the transcriptional regulation of GPR56 in different effector T cell populations has not yet been defined.

In a different disease setting, single-cell transcriptomic (scRNAseq) analysis of a tumor immune cell atlas dataset showed that GPR56 mRNA transcripts are expressed predominantly in CD8^+^ tumor-infiltrating lymphocytes (TILs) that displayed a (pre)exhausted and tumor-reactive phenotype [133]. As such, elevated GPR56 protein expression was detected in TILs of ovarian cancer patients and these GPR56-expressing TILs were mostly associated with the effector memory (T_EM_) and central memory (T_CM_) phenotypes. Finally, enforced GPR56 expression inhibited in vitro migration of primary T cells in response to SDF-1 [133]. Similarly, transcriptomic profiling of tumor-infiltrating neoantigen-reactive T cells of gastrointestinal cancer patients delineated an exhausted phenotype in the majority of CD8^+^ and CD4^+^ neoantigen-reactive TILs and showed that GPR56 was up-regulated significantly in CD4^+^ neoantigen-reactive TILs [134]. Despite its restricted expression in these effector T cell subsets, whether or how GPR56 modulate their immune effector functions awaits further studies. Taken together, GPR56 is expressed in diverse NK and T lymphocyte subsets with a functional link to various pathologies such as RA and cancers.

GPR56-collagen III interaction is critical for the proper lamination of the cerebral cortex, which is severely deviant in BFPP patients due to the loss of GPR56 [119]. More recently, Yeung et al. showed that this ligand-receptor pair also plays a role in hemostasis as GPR56 on platelets was found to promote the adhesion and activation of platelets in response to immobilized collagen in a shear force-dependent manner [135]. Indeed, GPR56-NTF was shed swiftly upon the interaction of platelets with immobilized collagen while under shear stress, leading to the CTF-mediated Gα_13_ signaling and eventually cell shape changes. Therefore, analogous to EMR2 and CD97, GPR56 is a novel mechanosensing receptor on platelets that responses to exposed collagen and blood flow.

During hematopoiesis, HSPCs in the mouse embryo and adult bone marrow express abundant GPR56, though the expression weakens significantly following cell differentiation [136]. Indeed, significant levels of GPR56 were detected in the CD34^−^c-Kit^+^Sca-1^+^Lin^−^ HSC fraction in which the GPR56^+^ HSC population contained the majority of the long-term repopulating (LTR) cells [137]. A role of GPR56 in the early stages of hematopoietic development was revealed by Kartalaei et al. who showed that GPR56 is involved in hematopoietic cluster formation during endothelial to hematopoietic cell transition in zebrafish and mouse [138]. Regardless, GPR56 is largely dispensable for steady state and regenerative hematopoiesis since the lack of GPR56 does not impair HSPC maintenance, migration, or function in steady-state or myeloablative stress-induced hematopoiesis [136]. Maglitto et al. later showed that this puzzled observation is likely due to the functional compensation of GPR56 by another ADGRG molecule, GPR97 [139].

In addition to HSCs/HSPCs, recent studies have uncovered the role of GPR56 as a surface marker and a disadvantageous prognostic factor of acute myeloid leukemia (AML) cells and leukemia stem cells (LSCs) [140,141,142,143,144]. High levels of GPR56 were expressed in EVI1^high^ (Ecotropic Viral Integration site-1) AML cells that showed a strong cell adhesion and anti-apoptotic phenotype [140]. Engraftment of GPR56-expressing HPCs significantly promoted myeloid leukemogenesis in mice, which was greatly impaired by receptor blockage mediated by GPR56-specific Ab [141]. Likewise, patients with GPR56^high^ AML cells at diagnosis had a higher relapse risk following allogeneic HSC transplantation [145]. These findings underline the importance of GPR56 in AML development and identify GPR56 as a potential therapeutic target of AML and LSCs.

In summary, GPR56 plays a role in the functional modulation of NK, T cells, and platelets, as well as leukemic cells.

## 7. ADGRG3/GPR97

### 7.1. Physiopathological Role of Murine ADGRG3/GPR97 in the Immune System

Murine Adgrg3/Gpr97, also named Pb99, was first cloned in 2000 and found to be expressed in pre-B cells and thymocytes, but not in mature B and T cells nor in nonlymphoid tissues. Initial investigation of *Pb99*-knock-out mice showed the receptor is dispensable for the normal development of B- and T-lymphocytes [146]. Nevertheless, Wang et al. generated another *Gpr97*-deficient mouse strain, which had significantly reduced mature B220^+^ lymphocytes but comparable immature B cell populations, suggesting a role for Gpr97 in the development of mature B cells in secondary lymphoid tissues [147]. Indeed, the spleen of *Gpr97*-deficient mice had a disorganized architecture of diffuse, irregular follicular B-cell areas and the disappearance of discrete marginal zones. Moreover, Ab responses to T cell-dependent and -independent Ags were enhanced and reduced, respectively, in the mutant mice likely via the modulation of the CREB and NF-κB signaling pathways. These findings uncovered a regulatory role of Gpr97 in B-lymphocyte fate decisions in mice [147].

The *Gpr97*-deficient mice were subsequently tested in various experimental disease models, revealing a role for Gpr97 in promoting obesity-associated macrophage inflammation, aggravating acute kidney injury (AKI), and reducing the severity of experimental autoimmune encephalomyelitis (EAE) [148,149,150]. On the other hand, Gpr97 is dispensable for inflammation in ovalbumin-induced allergic asthma and metabolic disorders in high-fat diet-induced obesity [148,151]. The molecular mechanisms whereby Gpr97 modulates mouse immune cell fate decisions and functions remain obscure.

### 7.2. Physiopathological Role of Human ADGRG3/GPR97 in the Immune System

Interestingly, unlike what was found in mice, extensive analyses of microarray transcriptomics of defined human leukocyte populations identified restricted GPR97 expression mostly in neutrophils and eosinophils, but not in lymphocytes [126]. We then confirmed and detected up-regulated GPR97 protein expression in blood and tissue-infiltrating neutrophils in systemic inflammation. Molecular and functional analyses indicated the role of GPR97 in promoting the antimicrobial activity of neutrophils [152]. More recently, we uncovered a novel GPR97-directed neutrophil activation mechanism involving the trans-activation of protease-activated receptor 2 (PAR2). Specifically, GPR97 interacted with membrane proteinase 3 (mPR3) on neutrophil surfaces and induced its allosteric activation. Augmented mPR3 then cleaved and activated PAR2, a GPCR renowned for its inflammatory function, leading to robust neutrophil activation (Figure 2). Intriguingly, GPR97-mediated PAR2 activation via mPR3 was strictly dependent on the formation of a macromolecular CD177/GPR97/PAR2/CD16b receptor complex. The role of the GPR97-associated receptor complex in pathologic inflammation was realized by upregulated GPR97 and mPR3 expression on the surface of neutrophils in various inflammatory diseases, including appendicitis, bacterial sepsis, and granulomatosis with polyangiitis (GPA). Our results hence reveal a novel aGPCR-GPCR transactivation mechanism in human neutrophils that directs inflammatory responses [153].

Altogether, GPR97 seems to be involved in different immune disorders in mice and humans because of its differential cellular expression patterns and distinct immune functions.

## 8. Concluding Remarks and Future Perspectives

Selective aGPCRs differentially modulate the development and function of HSPCs and various immune cell types, regulating innate and/or adaptive immune responses. Their uncontrolled expression and/or activity in turn leads to immune dysfunction and immune disorder. With the identification of specific interacting partners/ligands and the development of receptor-specific Abs and agonists/antagonists, it is now possible to delineate and manipulate the cellular functions of these aGPCRs in vitro and in vivo. Furthermore, the use of anti-CD97 and anti-EMR1 mAbs in animals has made it an attainable reality of treating immune disorders by aGPCR-targeting biologicals. In addition, gene-modified animals and diseased patients have provided invaluable avenues for functional validation. These efforts have established formally the functional importance of GPS proteolytic cleavage in the role of distinct aGPCRs as immune mechanosensors as exemplified by the involvement of EMR2 in VU, CD97 in cDC2 development, and GPR56 in platelet activation. Furthermore, the clear association of immune aGPCRs with various (auto)inflammatory/autoimmune pathologies and hematopoietic malignancies has marked these receptors attractive therapeutic targets of immune cell-related diseases.

In spite of these advances, many gaps in aGPCR immunobiology remain unfilled. To name a few, the specific signaling pathways of unique ligand-receptor pairs need to be defined more extensively as more diverse ligands are identified for a given aGPCR. For example, how does the same ELMO/Dock/Rac signaling axis downstream of BAI1 lead to bacteria-killing inflammatory and apoptotic cell-engulfing anti-inflammatory responses, respectively, in responding to LPS and PS? Another challenge is the functional redundancy of aGPCRs as reflected by the fact that many aGPCR knock-out animals did not show apparent immune phenotypes in steady states. This is likely due to shared ligands between homologous receptors and high degrees of similarity in receptor structure and expression profile. Noticeably, Maglitto et al. recently showed that Gpr56 and Gpr97 share functional redundancy in the development and differentiation of mouse hematopoietic cells [139]. Moreover, Chen et al. identified the essential amino acid L-Phe, derived from unique gut microbiota, as a bioactive agonist for both GPR56 and GPR97 [123]. These results indicate that some aGPCRs very likely cooperate and/or compensate each other functionally in various physiological and pathological systems. Application of single-cell technologies and spatial transcriptomics coupled with the latest structural insights of aGPCRs [154,155,156,157] shall provide a comprehensive understanding of aGPCR immunobiology and their roles in immunopathology.

## Figures and Tables

**Figure 1 ijms-24-05499-f001:**
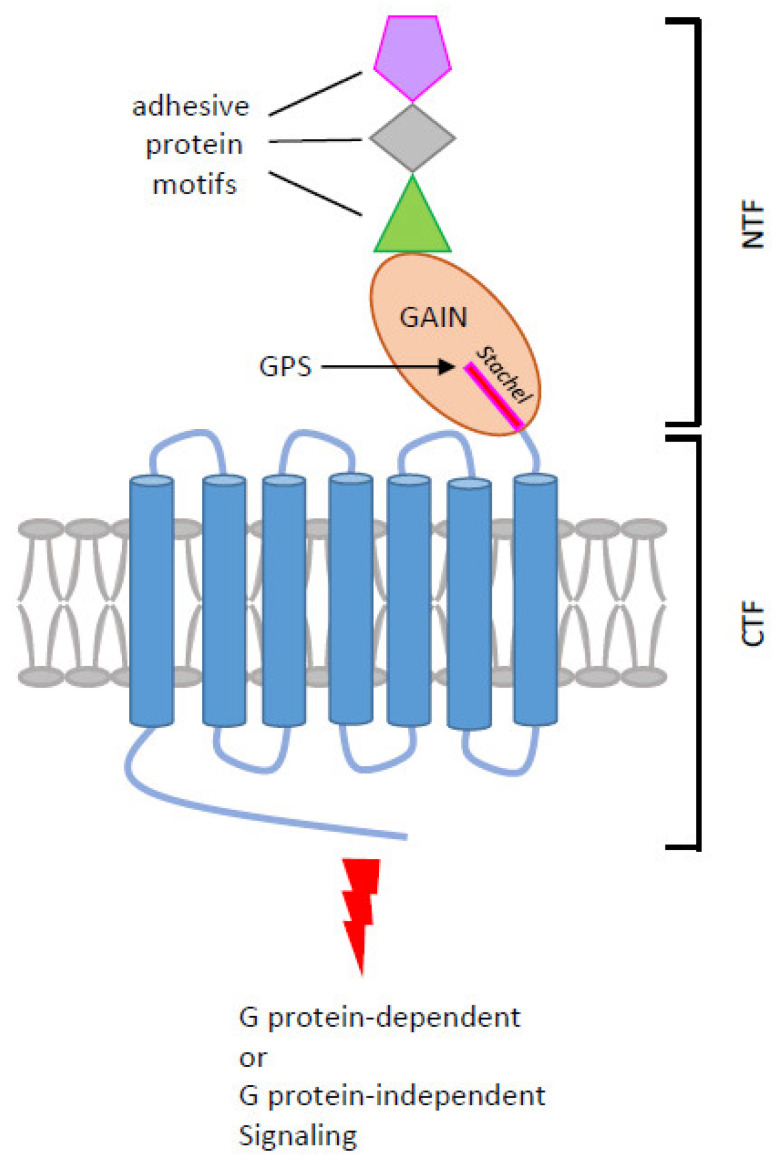
Structural and signaling characteristics of aGPCRs. The diagram shown is the schematic representation of aGPCR in general. The N-terminal fragment (NTF) contains various cell-adhesive protein motifs (colored shapes) followed by a GAIN domain in which an autoproteolytic reaction occurs at the consensus GPS motif. GPS autoproteolysis cleaves the receptor into the NTF and the C-terminal fragment (CTF) with a new N-terminal sequence (Stachel peptide). Upon ligand binding and/or mechanical stimulation, the NTF is dissociated/dislocated from the CTF to allow the Stachel peptide to bind to the 7TM region, inducing conformational changes and G protein-dependent or -independent signaling.

**Figure 2 ijms-24-05499-f002:**
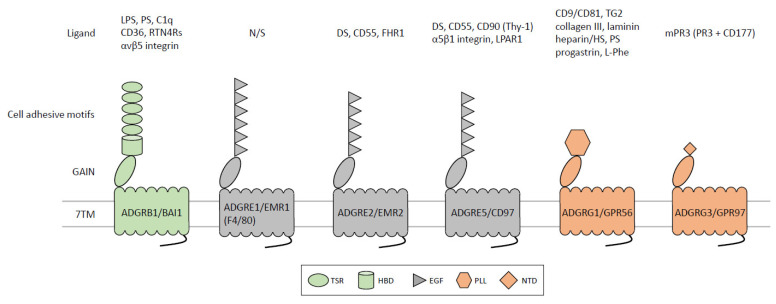
Molecular and functional characteristics of immune aGPCRs. The diagram shown is selective aGPCRs involved in immune dysfunctions. Different colored shapes in the N-terminal region represent various cell-adhesive protein motifs (detailed description in the lower panel). The GAIN domain is represented by a tilted oval. Ligands/interacting partners of each immune aGPCR are shown at the top panel. Abbreviations: C1q, complement component 1q; DS, dermatan sulfate; EGF, epidermal growth factor-like; FHR1, factor H-related protein 1; HBD, hormone-binding domain; HS, heparin sulfate; LPAR1, lysophosphatidic acid receptor 1; LPS, lipopolysaccharide; NTD, N-terminal domain; PLL, pentraxin/laminin/neurexin sex-hormone-binding globulin-like; PR3, proteinase 3; PS, phosphatidylserine; RTN4Rs, Reticulon-4 receptors; TG2, transglutaminase-2; TSR, thrombospondin type 1 repeat.

**Table 1 ijms-24-05499-t001:** Molecular characteristics and physiopathological roles of immune aGPCRs.

Receptor	Expression in Immune Cells	Ligand/Interacting Partner	Physiopathological Functions in the Immune System
ADGRB1/ BAI1	Macrophage, gastric phagocyte, microglia, T-ALL *	LPS, PS, C1q, CD36, αvβ5 integrin, RTN4Rs	Apoptotic cell clearance, colonic inflammatory disorders, HDL biogenesis, microglial neuronal removal, bacteria phagocytosis, anti-viral response, T-ALL pathogenesis.
ADGRE1/ EMR1 (Emr1, F4/80)	Monocyte, macrophage, eosinophil, and DC	N/S	Emr1 (F4/80): Anti-intracellular bacterial infection, induction of Ag-specific peripheral immune tolerance. EMR1: homeostatic control of human eosinophils.
ADGRE2/ EMR2	Monocyte, macrophage, granulocyte, mast cell, and DC	DS, CD55, FHR1	Neutrophil activation, differentiation and inflammatory activation of macrophage, NLRP3 inflammasome activation, vibration-induced mast cell activation, and degranulation (VU, hereditary α-tryptasemia).
ADGRE5/ CD97	T and B lymphocytes, monocyte, macrophage, granulocyte, DC, HSPCs, AML, BL, LSC	DS, CD55, CD90 (Thy-1), α5β1 integrin, LPAR1	Leukocyte adhesion/trafficking, homeostasis and anti-bacterial response of granulocytes, activation, and expansion of T cell subsets, homeostasis and Agpresenting functions of splenic cDC2s, IS stability, regulation of hematopoiesis and LSC maintenance.
ADGRG1/ GPR56	NK, cytotoxic T cells, T_PH_ and Treg cells, TILs, platelet, HSPCs, AML, LSC	CD9/CD81, TG2, collagen III, laminin, heparin/HS, progastrin, L-Phe, PS	NK inhibitory receptor, up-regulated expression in T cell subsets of RA, JIA, and various cancer types, platelet activation in response to exposed collagen and blood flow, hematopoietic development, tumorigenesis of AML.
ADGRG3/ GPR97	Gpr97: pre-B cells and thymocytes. GPR97: neutrophil and eosinophil	N/A mPR3 (PR3 + CD177)	Mouse: B-lymphocyte fate decision, obesity-associated macrophage inflammation, pathogenic modulation of AKI and EAE. Human: inflammatory activation of neutrophilic granulocytes.

* Abbreviations: T-ALL, T-cell acute lymphoblastic leukemia; RTN4Rs, Reticulon-4 receptors; DS, dermatan sulfate; FHR1, factor H-related protein 1; VU, vibratory urticarial; HSPCs, hematopoietic stem and progenitor cells; AML, acute myeloid leukemia; BL, Burkitt lymphoma; LSC, leukemic stem cell; LPAR1, lysophosphatidic acid receptor 1; IS, immunological synapse; TG2, transglutaminase-2; HS, heparin sulfate; PS, phosphatidylserine; AKI, acute kidney injury; EAE, experimental autoimmune encephalomyelitis.

## Data Availability

Not applicable.

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
