# Peer review of "Role of Adhesion G Protein-Coupled Receptors in Immune Dysfunction and Disorder"

_ijms, 2023, doi:10.3390/ijms24065499_

Round 1
Reviewer 1 Report
The authors aimed to discuss the molecular and functional characteristics of distinctive immune aGPCRs and their physiopathological roles in the immune system. Several main points need to be improved.
1) Figure 1 and Table 1 were absent. It needs to be resubmitted and evaluated;
2) The text description about each part (2-7) of selective members of the ADGRB, ADGRE, and ADGRG subfamily present in this paper seemed to be very confused for the readers. It needs to be reorganized. In each part, subtitles using are recommended to make the readers better understanding about what the authors really aim to talk and focus on here.
3) Disorders of the immune system including immunodeficiency; immuno-malignancy; and (auto)inflammatory; autoimmune; and allergic diseases (Abstract section indicated). Are they all involved in each of the parts? Or different aGPCRs member has distinctive immune roles? It requires clearer definition in each part (2-7).
4) In the part (2-7), for better clearer understanding of the molecular characteristics of GPCRs in the immune system, 2-3 figures about molecular pattern diagrams are suggested.
Reviewer 2 Report
In this review submitted to IJMS the authors effort to collect information for the most relevant aGPCRs implicated in immune disorders, like BAI1, since these can be therapeutic targets of great interest.
I think this review gives a good summary of the current scientific landscape of aGPCRs. However, I think the information should be represented visually with more figures/tables to allow access to the content at a glance.
Detected issues:
- Figure 1 was referenced in introduction, however, is not included in the manuscript. Neither Table 1.
- Introduction seems to have different typographies.
Reviewer 3 Report
In the manuscript entitled “Role of Adhesion G Protein-Coupled Receptors in Immune Dysfunction and Disorder”, Tseng et al. summarize the current knowledge of the role played by aGPCRs in innate and adaptative immunity as well as in autoimmune diseases. Although several aspects of aGPCR biology in inflammation and immunity have been elucidated many others remain elusive prompting to continue and extend the investigations on aGPCR role in physiological and pathological conditions. I would like to mention that aGPCR biology and emerging role in pathophysiology have been extensively reviewed in a recent paper by Lala and Hall (Physiol Rev 102: 1587–1624, 2022; doi:10.1152/physrev.00027.2021).
Round 2
Reviewer 1 Report
The authors have answered the questions. But what's problem, I can not see any Figures included this paper.
Author Response
We thank the reviewer for the comment. The revised manuscript with new figures was re-submitted as requested in the previous submission. It is not known to us why the figures and Table were lost. I attach here the pdf file of the original merged manuscript for your attention.
